# Flavolignans from Silymarin as Nrf2 Bioactivators and Their Therapeutic Applications

**DOI:** 10.3390/biomedicines8050122

**Published:** 2020-05-14

**Authors:** Nancy Vargas-Mendoza, Ángel Morales-González, Mauricio Morales-Martínez, Marvin A. Soriano-Ursúa, Luis Delgado-Olivares, Eli Mireya Sandoval-Gallegos, Eduardo Madrigal-Bujaidar, Isela Álvarez-González, Eduardo Madrigal-Santillán, José A. Morales-Gonzalez

**Affiliations:** 1Laboratorio de Medicina de Conservación, Escuela Superior de Medicina, Instituto Politécnico Nacional, México Escuela Superior de Medicina, Plan de San Luis y Díaz Mirón, Col. Casco de Santo Tomás, Alcaldía Miguel Hidalgo, Mexico City 11340, Mexico; nvargas_mendoza@hotmail.com; 2Escuela Superior de Cómputo, Instituto Politécnico Nacional, Av. Juan de Dios Bátiz s/n esquina Miguel Othón de Mendizabal, Unidad Profesional Adolfo López Mateos, Mexico City CP 07738, Mexico; anmorales@ipn.mx; 3Licenciatura en Nutrición, Universidad Intercontinental, Insurgentes Sur 4303, Santa Úrsula Xitla, Alcaldía Tlalpan, Mexico City CP 14420, Mexico; mtz98mauxd@gmail.com; 4Academia de Fisiología Humana, Escuela Superior de Medicina, Instituto Politécnico Nacional, Plan de San Luis y Díaz Mirón, Col. Casco de Santo Tomás, Del. Miguel Hidalgo, Mexico City 11340, Mexico; soum13mx@gmail.com; 5Centro de Investigación Interdisciplinario, Área Académica de Nutrición, Instituto de Ciencias de la Salud. Universidad Autónoma del Estado de Hidalgo. Circuito Actopan-Tilcuauttla, s/n, Ex hacienda La Concepción, San Agustín Tlaxiaca, Hidalgo CP 42160, Mexico; ldelgado@uaeh.edu.mx (L.D.-O.); eli_sandoval7987@uaeh.edu.mx (E.M.S.-G.); 6Escuela Nacional de Ciencias Biológicas, Instituto Politécnico Nacional, “Unidad Profesional A. López Mateos”. Av. Wilfrido Massieu. Col., Zacatenco, Mexico City 07738, Mexico; eduardo.madrigal@lycos.com (E.M.-B.); isela.alvarez@gmail.com (I.Á.-G.)

**Keywords:** silymarin, flavolignans, Nrf2, antioxidants, bioactivators

## Abstract

Silymarin (SM) is a mixture of flavolignans extracted from the seeds of species derived from *Silybum marianum*, commonly known as milk thistle or St. Mary’sthistle. These species have been widely used in the treatment of liver disorders in traditional medicine since ancient times. Several properties had been attributed to the major SM flavolignans components, identified as silybin, isosilybin, silychristin, isosilychristin, and silydianin. Previous research reported antioxidant and protective activities, which are probably related to the activation of the nuclear factor erythroid 2 (NFE2)-related factor 2 (Nrf2), known as a master regulator of the cytoprotector response. Nrf2 is a redox-sensitive nuclear transcription factor able to induce the downstream-associated genes. The disruption of Nrf2 signaling has been associated with different pathological conditions. Some identified phytochemicals from SM had shown to participate in the Nrf2 signaling pathway; in particular, they have been suggested as activators that disrupt interactions in the Keap1-Nrf2 system, but also as antioxidants or with additional actions regarding Nrf2 regulation. Thus, the study of these molecules makes them appear attractive as novel targets for the treatment or prevention of several diseases.

## 1. Introduction

Living organisms are constantly exposed to different insults from the environment or as a result of biochemical reactions. Aerobic metabolism demands a constant oxygen flux for energy production, but the cost is the potential synthesis of reactive oxygen species (ROS) such as superoxide anion radical (^•^O_2_^−^), hydroxyl radical (^•^OH^‾^), and hydrogen peroxide (H_2_O_2_) [1]. These highly reactive molecules possess the ability to bond with other cellular structures such as lipids, proteins, and DNA, increasing oxidative stress, causing damage. ROS are generated basically in mitochondria, peroxisomes, endoplasmic reticulum, in nicotinamide adenine dinucleotide phosphate oxidase (NADPH oxidase), and in electron transfer enzymes such as, for instance, cytochrome P450 (CYP450) isoforms [2]. Mitochondria comprise the main ROS producer; during electron transport, molecular oxygen is reduced by cytochrome c oxidase into H_2_O and, in turn, the incomplete reduction of oxygen leads the genesis of ROS [3]. It is noteworthy that ROS are essential for maintaining homeostasis, contrary to it being thought that they are regarded as harmful. ROS make possible some signaling pathways: cell proliferation, differentiation, stress, and metabolic adaptation are mediated by ROS [4,5]. In addition, ROS, as electrophilic molecules, can modulate receptor tyrosine kinase signaling [6], regulate protein activities by oxidation of the thiol (–SH) moiety, and the optimal tyrosine phosphorylation of protein-tyrosine phosphatase (PTP) 1B in insulin signaling, among others [7]. Therefore, it is crucial to maintain equilibrium between ROS production and elimination for homeostasis. In this regard, cells have developed a potent powerful system to protect themselves from damage. Nuclear factor erythroid 2 (NFE2)-related factor 2 (Nrf2) is known as a master regulator of antioxidant response; its activation is mediated by ROS [8]. The tremendous impact observed in antioxidant response generated by the activation of Nrf2 has promoted the carrying out of many investigations in order to understand the mechanisms implicated in their induction. In addition to ROS, other factors have been identified as Nrf2 inducers, such as several phytochemicals, also denominated bioactivators [9], electrophilic molecules, and physical exercise [10]. 

The impaired Nrf2 pathway is associated with aging, inflammation, organ and tissue damage, and the consequent development of chronic diseases [11]. Thus, the objective of this work was to review the role of SM and the flavolingans as potent inducers of Nrf2. According to previous reports, SM has been widely used in traditional medicine in the treatment of hepatic disorders for many years, but formal studies have demonstrated that it is able to induce an antioxidant response [12]. The mechanisms are not completely understood; however, SM or its flavolignans appear to be a promising target for further investigations related to inducing the Nrf2 signaling pathway as a proposal in the treatment of several diseases. 

## 2. The Signaling Pathway Nrf2/Keap1/ARE as Cytoprotector System

The cell is exposed to different environmental insults that are related to damage; therefore, it must develop an effective system to protect itself. These systems are related to the activation of detoxifying and maintaining redox homeostasis pathways. Nrf2 is a ubiquitous member of the basic leucine zipper (bZIP) transcription factor family, encoded by the *NFE2L2* gene that regulates a battery of approximately 250 genes involved in multiple cellular processes; cytoprotection, the redox response, and detoxification are the most important of these [13]. Nrf2 is called the master regulator of stress responses. However, it is also linked to cell proliferation and differentiation, growth, apoptosis, and carbohydrate and lipid metabolism [14]. It was first described by Yuetwai Kan and identified as a 66 kDa protein that contains a C-terminus Cap’N’Collar basic-region leucine ZIPper (CNC-bZIP) domain with high homology to nuclear factor erythroid 2 (NF-E2) [15]. Thus, it was named Nrf2 and recognized as the third member of the CNC-bZIP family in mammals [16]. Nfr2 was initially discovered as a transcriptional regulator of β-globin genes [16], and it was hypothesized as a fundamental factor for hematopoiesis. Soon after this, Nrf2-induced antioxidant response element (ARE)-mediated genes were found [17].

In humans, Nrf2 is a modular protein of 605 amino acids divided into different regions called Nrf2-ECH homology (Neh) domains; seven domains (Neh1-7) have been identified, with diverse functions (Figure 1) [18]. The Neh1 domain comprises the conserved CNC-bZIP region that is fundamental for heterodimerization with other bZIP proteins as a transcription factor [19]. The N-terminal Neh2 contains a pair of highly conserved peptide sequences to which Keap1 binds; these comprise the high-affinity ETGE motif and the lower-affinity DLG motif [20]. C-terminal Neh3 interacts with co-activator CHD, the transactivation responsible for ARE-dependent genes after chromatin remodeling. This co-activator is a chromo ATPase/helicase DNA-binding protein [21]; Neh4 and Neh5 are domains of transcription activation that bind to the adenosine monophosphate (AMP)-responsive element-binding protein. In addition, these can also interact with nuclear factor RAC3/AIB1/S, enhancing Nrf2-target ARE gene expression [18,22]. Contrariwise, the Neh7 domain can repress Nrf2 by interacting with the retinoic X receptor [23].

### Transcription, Regulation and the Antioxidant Response

Under homeostatic conditions, Nrf2 remains in the cytoplasm at an overall low concentration in the cell. It remains inactive bonding to its negative regulayor Kelch-like ECH-associated protein (Keap1) through the interaction of the Neh2 domain [24]. Keap1 is a 624 amino-acid protein redox regulator substrate adaptor for the Cullin (Cul)3-RING-box protein (Rbx)1 ubiquitin ligase, directing Nrf2 toward its degradation by the ubiquitination complex. There are two binding sites in the Neh2 domain: ETGE and DLG motifs, which form a dimer link with Keap1, leading to proteosomal degradation. This two-binding-site model requires the ubiquitination of seven lysines [25]. The ubiquitination of Nrf2 leads to rapid degradation by the 26S proteosome, maintaining it at low levels in the cell [26]. The increase of oxidative stress by electrophils/ROS or in the presence of activating compounds caused the oxidation of cysteine residues in Keap1, a rich-cysteine protein, producing a conformational change. Thus, Nrf2 is released from Keap1, accumulating in and translocating into the nucleus, where it binds to small musculo-aponeurotic fibrosarcoma proteins (sMaf) in a specific DNA sequence: the antioxidant response element (ARE) [8]. When it is active, Nrf2 transcriptionally upregulates the cytoprotective system. ARE were identified as a core DNA sequence 5′-puGTGACNNNGC-3′ [27] by the induction of phase-II drug metabolism enzymes in the presence of phenolic compounds [28]. The Nrf2/sMaf/ARE complex plays physiological roles in detoxification, the anti-inflammatory process, autophages, and proteosomes. Later, Nrf2 is driven outside the nucleus immediately after the occurrence of the stress situation as a result of the nuclear export signal (NES), ending the antioxidant response signal [29]. The Nrf2/Keap1/ARE pathway must be modulated in terms of both activation and inhibition in order to preserve equilibrium. Consequently, the disruption of Nrf2/Keap1 activates the pathway, but the activation of the cyclin-dependent kinase inhibitor p21Cip/WAF1 binds to the DLG motif in the Nrf2 and competes with Keap1, avoiding Nrf2 ubiquitination [30]. On the other hand, the increase in oxidative stress levels leads to the activation of p62 [31], a ubiquitin-binding protein that triggers degradation through the lysosome or proteosome pathway. P62 activates Nrf2 by the sequestration of Keap1, and is degraded by autophagy under homeostatic conditions [32,33].

Since the discovery of ARE, numerous investigations have resulted in order to describe the regulatory mechanisms of the Nrf2/ARE activation pathway [34]. Many natural or synthetic compounds have proven to be able to induce Nrf2, such as polyphenols, allyl sulfides, flavonoids, isothiocyanates, dithiolethiones, and triterpenoids, and it has been proposed for use in the prevention or treatment of a range of diseases related with oxidative stress (Figure 2) [8]. Some of these will be discussed further. 

Nrf2 is known as the master cytoprotector regulator of the antioxidant response, and the target genes activated by Nrf2 exhibit detoxification and antioxidant properties against oxidative stress and xenobiotics. The synthesis of glutathione (GHS) is one of the most important activities. GHS is a tripeptide formed from glutamate, cysteine, and glycine, and is considered the main non-enzymatic endogenous antioxidant. In its reduced form, GHS is capable of donating electrons to reduce the disulfur links of cysteine amino acids in cytoplasmic proteins; during this process, it is converted into its oxidized form: GSSG [35]. In GSH biosynthesis and metabolism, there is the involvement of a great variety of enzymes, such as the glutamate-cysteíne ligase (GCL) complex in the binding of the glutamate and cysteine amino acids. The latter are made up of the glutamate-cysteine ligase modifier subunit (GCLM) and the glutamate-cysteine ligase catalytic subunit (GCLC), including the enzymes glutathione *S*-transferase (GST), glutathione peroxidase (GPx), glutathione reductase (GR), and the XCT protein, which transports cysteine into the interior of the cell to be reduced into cysteine from GSH [36], in turn supplying the demand of cysteine for the biosynthesis of GSH [37]. 

Nfr2 also regulates the expression of some phase-II detoxifying enzymes in the metabolism of xenobiotics. These are fundamentally enzymes that are hydrophilic to metabolic groups in order to render them more soluble and to facilitate their elimination [38]. Among these are included the UDP-glucuronosyl transferases (UGT), the sulfotransferases (SULT), and heme-oxygenase1 (HO-1) [39]. The antioxidant protector system also includes the production of the NADPH required for the multiple redox reactions of the system itself. The enzymes glucose-6-phosphate dehydrogenase (G6PD), isocytrate dehydrogenase (IDH), and malic enzyme 1 (ME1) are responsible for the formation of NADPH in different metabolic pathways. On the other hand, thioredoxin 1 (TXN1) [40] and thioredoxin reductase 1 (TXNRD1) can oxidize the thiols of proteins, for example, the thiols of DNA repair protein Ref1. Finally, the quinones, on being metabolized, produce a great amount of ROS; for this, Nrf2 regulates the expression of NAD(P)H-quinone oxidoreductase-1 (NQO1) and aldo-keto reductase (AKR) for converting quinones into less toxic products [41]. 

## 3. Nrf2 as a Target for Therapeutic Models

In addition to the antioxidant and detoxifying systems, Nrf2 also participates in maintaining physiological conditions, in order for there to be a continuation of homeostasis involving cell proliferation, cell differentiation, the anti-inflammatory and the aging processes, among others. In that diverse studies demonstrate that the abrogation of Nrf2 directly affects cell proliferation in mouse embryotic fibroblasts and alveolar type- II cells related to GSH depletion by downregulation of GCLM, GCLC, TXN1, TXNRD1, and PRDX1 [42,43]. The loss of Nrf2 is linked to impairment of the ability of scavenging ROS in cardiomyocytes, bone marrow-derived macrophages, and dendritic cells, rendering them more susceptible to damage by oxidative stress [44,45]. The process of mitosis is also affected by the Nrf2 abrogation observed in the G2/M-phase arrest of alveolar epithelial cells due to the lack of cyclin B1 and the cyclin-dependent kinase 1 (CDK1) plus the reduction of the phosphorylation of retinoblastoma protein in Ser-807 and Ser-811 [46]. Nrf2 has additionally been associated with differentiation from basal stem cells; thus, in hematopoietic cells, it positively regulates CXCR4, supporting the cells forming in the bone marrow [47].

### 3.1. Nrf2 in Aging and Muscular Dysfunction 

One of the main alterations associated with an increase in age is muscular atrophy and diminution in the functions of skeletal-muscle mobility and contractility [48]. This reduction in muscular mass initiates in the fourth decade of life; a loss of around 8% occurs at up to 70 years of age; from that point, the loss increases to 15% per decade. There are changes in body composition, a biological diminution in the metabolic rate that conditions changes such as the imminent weight gain of body fat and the reduction of the fat-free mass proper to advanced age [49].

Aging determines a diminution in the capacity of the cellular and molecular systems to mediate oxidative stress, augmenting the level of ROS and reactive nitrogen species (RNS), producing mitochondrial damage and damage to other subcellular structures [50]. One of the signs of cellular aging manifests as the detriment to the functions of damage repair in mitochondrial DNA (mtDNA) [51,52] under certain conditions, such as the energy supply and signaling caused by high ROS levels [53]. Consequently, the mitochondrial conditions raise, to an even greater degree, the amount of ROS that can activate and induce death, reducing the number of muscle fibers [54]. Previous studies have shown that muscular atrophy and its extension is related to the increase in oxidative stress due to the mitochondrial production of ROS [55,56]. Even more so, a direct relationship has been observed between the absence of the CuZnSOD1 enzyme and the increase of oxidative stress, mitochondrial damage, diminution of contractility, and muscular atrophy [57,58]. To this, we must add lack of physical activity, eating disorders, associated diseases and, in general, poorly healthy lifestyles, factors that accelerate the loss of the fat-free mass [59]. A recently published study [60] mentioned that mice with mitochondria lacking skeletal muscle superoxide dismutase 2 (*mSod2*KO) resulted in high levels of mitochondrial ROS, measured by the increase in damage markers, such as lipoperoxidation, glutathione oxidation, modification in the oxidation of myofibrillar proteins, and the increase of superoxide in the mitochondrial complexes, generating the dysfunction of succionate dehydrogenase. This severely affected the production of ATP and tolerance for exercise. Nonetheless, and contrary to what was expected, there was an increase of muscle mass reflected in an increase of the branching of fibers and nuclei. Similarly, a reduction was observed of the levels of intracellular calcium, which was associated with diminution of the contraction force. These results suggest that the elevation of mitochondrial oxidative stress generates neuromuscular disruption and contraction, but not atrophy, probably due to the cellular capacity to mediate stress through the formation of fibrillary branching. 

The work of Kitaoka et al. [61] indicates that markers of mitochondrial oxidative stress were significantly high in *Nrf2* Knock Out (*Nrf2*KO) mice; there was a decline of mitochondrial respiration regardless of age in *Nrf2*KO as well as in Wild-Type (WT) mice. Additionally, less activity of cytochrome oxidase was observed in *Nrf2*KO as well as in WT mice of advanced age compared with young mice, accompanied by a high production of mitochondrial ROS in *Nrf2*KO mice, and there was no affectation of the amount of muscular mass relative to weight, indicating that the deficiency of Nrf2 is not directly related with the decline of the respiratory function of skeletal muscle. Nonetheless, there was an increase of mitochondrial oxidative stress related to age. The most evident morphofunctional conditions take place in type-II muscle fibers, probably due to that they are the motor units of the greatest size utilized in high-powered physical work, explosivity, and short duration. These possess a low oxidative capacity, thus a lower number of mitochondria, while type-I muscle fibers, which are smaller in size and elongated, possess greater mitochondrial density due to their high oxidative capacity and are predominantly recruited in works long in duration and of moderate intensity [62]. Thus, senescence of the type-II fibers favors muscular atrophy, a phenomenon that can be estimated through the progressive reduction in muscular volume that characterizes older adults.

Diminution in muscular mass and its functionality have also been linked with down expression of Nrf2 [11]. In wild-type (WT) and in *Nrf2−/−* young (4 months of age) and old (24 months of age) mice, a reduction of muscle mass and in contraction capacity has been observed in old mice with the *Nrf2−/−* genotype compared with those of the same age. This was associated with oxygen consumption, the greater mitochondrial production of ROS, redox imbalance, an increase in the nitrosylation of proteins, and a reduction in the expression of the acetylcholine receptor, suggesting that the expression of Nrf2 plays a fundamental role in preventing the development of sarcopenia [63]. It was previously sustained that disruption in the *Nrf2*KO gene in young mice (2 months of age) entertains minimal implications in antioxidant defenses, although there is a diminution in the messenger RNA (mRNA) of the NQO1 protein compared with that of WT mice. In contrast, in old mice (24 months of age), there is a significant increase of ROS with a detriment to GSH, leading to the increase of oxidative stress, ubiquitination, and pro-apoptotic signals as compared with WT of the same age. This suggests that the antioxidant response in muscle is severely compromised with age [64].

### 3.2. Nrf2 and Inflammation

Increasing evidence had proposed that continuous oxidative stress leads to chronic inflammation, which is one of the main causes of chronic diseases. Inflammation produces ROS and other electrophilic molecules, inducing lipid peroxidation and protein and DNA damage. In some way, Nrf2 controls inflammation and mediates the magnitude of the inflammatory response, because it is known that it upregulates the proliferator-activated receptor-γ (PPARγ), which triggers the anti-inflammatory response. Certain levels of ROS are needed to activate the signaling of proliferation, differentiation, and adaptation; however, abnormal ROS production induces nuclear factor-kappa beta (NF-κB) [65,66]. NF-κB can be activated by the ablation of Nrf2 as a result of the high level of ROS, promoting signaling via the nonreceptor proto-oncogene tyrosine-protein kinase c-Src, Abelson murine leukemia viral oncogene homolog 1 (c-Abl), protein kinase C (PKC)δ, and protein kinase D (PKD) [67,68].

The activation of the Nrf2/ARE signaling pathway represses the activation of proinflammatory genes and anti-inflammatory pathways (Figure 3). The inflammation response is in part regulated by cytoprotector genes such as *GPx* and *TXN*, and there is a positive relation in the reduction of TNF-α- and IL-3-induced inflammation and the overexpression of HO-1 via the suppression of activator protein-1 (AP-1)-binding-DNA [69]. At the same time, HO-1 downregulates the expression of TNF-α and IL-Β through the inhibition of the lipopolysaccharide (LPS)-impairing inflammatory response [70]. Nrf2 interferes with LPS-induced transcriptional regulation of proinflammatory cytokines IL-6 and IL-β in macrophages, inhibiting RNA polymerase II recruitment by binding to the proximity of the corresponding genes [71]. Recently, the Toll-like receptors (TLR) signaling pathway has been associated with immune responses via the production of inflammatory cytokines, TNF-α and IL-6, chemokines, the macrophage inflammatory protein 2 (MIP2), and IL-8 and Interferon type I. In addition, TLR have been associated with Nrf2 crosstalks in the reduction of TLR-driven inflammation by the action of different kinases (protein kinase C, Burton’s tyrosine kinase, and MAPK), and also by the inhibition of IL-6, IL-β, and TNFα, throughout the expression of cytoprotectors HO-1, NQO1, and SOD [72,73] and finally, by p62-mediated autophagy [74]. 

### 3.3. Nrf2 and Chronic Diseases 

It is well-supported that oxidative stress plays a crucial role in developing pathological processes such as, for instance, cancer, diabetes, neurodegenerative diseases, pulmonary disorders, liver and renal dysfunctions, and cardiovascular diseases. Since the discovery of Nrf2, a large number of investigations have emerged to elucidate its participation in chronic diseases [75].

#### 3.3.1. Nrf2 in Cancer

Notwithstanding the widely supported evidence of Nrf2 cytoprotection, recently accumulated evidence suggests that Nrf2 plays a paradoxical role in cancer. On the one hand, there is sufficient research reporting the ability of Nrf2 to suppress carcinogenesis at early stages, owing to the maintenance of cellular redox homeostasis, the latter conferring an increased detoxification ability that might protect it, ensuring cell survival. On the other hand, the overactivation of Nrf2 in diverse tumors induces pro-survival genes, promoting cancer-cell proliferation, the repression of apoptosis, and the greater capacity of the self-renewal of cancer stem cells [76]. Alterations in the Nrf2/Keap1 pathway are one of the foremost causes of oncogenic activation. Mutations in key genes, genetic changes such as copy-number variations, or diverse polymorphisms (single nucleotide polymorphisms [SNP]) are responsible for aberrant Nrf2/Keap1 signaling [77].

It has been thought that Nrf2-deficient mice are more susceptible to suffering damage via oxidative stress as a result of the impairment of antioxidant defenses to fight against exogenous insults, leading to DNA damage and tumorigenesis [78]. Nonetheless, Nrf2 hyperactivity confers on cancer cells the characteristics of rapid and infinite division, growth, and proliferation, skipping apoptosis, the induction of angiogenesis, metastasis, and resistance to therapy [76]. It was noted previously that Nrf2 regulates the expression of metabolic enzymes such as G6PD, PGD, and TKT, as well as of the enzymes involved in NADPH synthesis; with regard to cancer cells, the overactivation of Nrf2 implies the induction of these enzymes, ensuring the availability of energy for cell division and growth and contributing to metabolic reprogramming for cell proliferation [79]. The deficiency of Nrf2 induces cell-cycle arrest in the G2/M phase [80]; in addition, it is involved directly in cell proliferation because it interferes in cell-cycle regulation due to its target-associated genes *Pdgf-c*, *Igf1*, *Itgb2*, *Jag 1*, and *Bmpr1a* [81]. A deficit of Nrf2 affects the epidermal growth factor receptor (EGFR) signaling pathway, leading to impaired mRNA translation in pancreatic cells [82].

Cancer cells are characterized by their escape from apoptosis. Under normal stress conditions, ROS induced antioxidant enzymes, but the superfluous levels of ROS may excessively activate Nrf2, their becoming resistant to cell death. An interaction has been identified as existing between Nrf2 and other signaling apoptosis mediators, such as B-cell lymphoma 2 (Bcl-2), an antiapoptotic protein that promotes cell survival [83]. Activation of the proapoptotic c-Jun N-terminal kinases (JNK) is shut down by the Nrf2 downstream gene glutathione-S-transferase pi 1 (*GSTP1*) [84]. Together with that, cancer stem cells (CSC) possess a self-renewal capacity that confers on them the ability to proliferate rapidly, rendering cells more resistant to therapy and a greater possibility of tumor relapse after treatment. These qualities are possibly due to a higher DNA repair capacity, a prominent expression of antioxidant defenses, and to a better drug tolerance [85,86]. To conclude, apoptosis is decreased by the overactivation of Nrf2 induced by high oxidative stress levels [87]. 

Nrf2 induces angiogenesis through the expression of HO-1, linked to the induction of the vascular endothelial growth factor (VEGF), promoting proliferation, migration, and the formation of new capillaries. Hypoxia-inducible factor 1α (HIF-1α) showed a close relationship with VEGF signaling. Nrf2 blockaded the activation of HIF-1α/VEFG signaling, suppressing angiogenesis in tumor [88]. On the other hand, an aberrant accumulation has been observed of p62 in certain types of cancer. Nrf2 and p62 are independent prognostic factors for non-small cell lung cancer (SCLC) in patients with adenocarcinoma, suggesting that molecular mechanisms in the evolution of cancer are different in adenocarcinoma- and squamous cell carcinoma [89].

#### 3.3.2. Nrf2 in Diabetes

Diabetes has been related with an excessive production of ROS and RNS, as a result of chronic hyperglycemia, which leads to an unbalanced redox dysfunction with the development of complications, specifically kidney failure, microvascular changes, peripheral neuropathy, and retinal damage [90]. Alternatively, greater damage has been reported in terms of neuropathy, nephropathy [91], cardiomyopathy, and retinopathy [92] in *Nrf2*KO mice compared with their counterpart *Nrf2*WT in diabetic rodent models.

The increase of ROS levels interferes as signaling molecules in pancreatic β-cells by the secretion of glucose-stimulated insulin. In this respect, the lack of Nrf2 has shown alterations in insulin sensitivity. One of the possible mechanisms is due to that protein kinase B/Akt (Akt) displayed greater phosphorylation at Ser-473 of the skeletal muscle and livers of *Nrf2*KO mice. Other signaling pathways are implicated in diabetes, such as the mammalian target of Rapamycin (mTOR) in mTOR complex 1 (mTORC1) and mTOR complex 2 (mTORC2), which appear to be deregulated in the development and progression of diabetes [93]. It has been shown that the loss of insulin sensitivity and glucose intolerance is caused by the inhibition of transcription factor p70^S6K^, a target of mTORC1 [94]. In addition, high levels of glucose trigger the regulation of mTORC2 signaling, affecting Akt/GSK-3 signaling and inducing changes in Nrf2 protein stability [94]. Nrf2 modulates AMPK phosphorylation, improving glucose sensitivity and insulin resistance [95]. Previous studies indicated that Nrf2 regulates the antioxidant enzyme system in pancreatic β-cells [96], contributes to protection against inflammation, and regulates autophagy and the expression of proteosome catalytic subunits in β-cells [97,98]. Moreover, in skeletal muscle (SkM)-specific Keap1 Knock-Out (*Keap1Mu*KO) mice, mice that expressed abundant Nrf2 and blood glucose levels were significantly downregulated, as were the levels of muscle-type *PhKα* subunit (*Phkα*), while glycogen branching enzyme (*Gbe1*) mRNA, along with glycogen branching enzyme (GBE) and phosphorylase b kinase α subunit (Phkα) protein, were upregulated in mouse SkM. Additionally, glucose uptake was improved due to the reduction of the glycogen content, forcing GBE expression in C2C12 myotubes. In conclusion, Nrf2 induction increases glucose tolerance because it regulates glycogen metabolism in SkM and liver [99].

#### 3.3.3. Nrf2 in Neurodegenerative Diseases 

The protective role of Nrf2 in neurodegenerative disorders due to the increase of oxidative stress is being widely studied. The absence of Nrf2 produces a greater loss of dopaminergic neurons in the substantia nigra, inflammation, and severe astrogliosis and microgliosis in a Parkinson’s disease (PD) rodent model induced by 1-methy-4-phenyl-1,2,3,6-tetrahydropyridine (MPTP), a potent neurotoxin. Nonetheless, the overexpression of Nrt2 protects astrocytes against MPTP toxicity, suggesting that modulation of the Nrf2 pathway could be a potential target for PD [100]. It was shown that a functional haplotype in the *NLFE2L2* gene promoter of Nrf2 is related to reducing the risk ofhaving PD [101]. Mitochondrial dysfunction is one of the crucial ethological factors in PD; there is recent evidence that reveals different types of crosstalk between mitochondria and the Keap1/Nrf2 pathway [102]. Endothelial function is affected by the absence of Nrf2, as well as size, in cerebral infarct and neurological damage after an ischemic event. Endothelial function is reduced in Nrf2-deficient models [103].

Age is also associated with neurodegeneration, with mitochondrial disruption and the defeat of membrane potential. Activation of the immune cells is the main cause of neuronal damage in multiple sclerosis; then, in *Nrf2*KO mice, there is marked overactivation and an aggressive response against the central nervous system (CNS) plus neuronal demyelation compared with WT [104]. Alzheimer’s disease (AD) is probably the most important neurodegenerative disorder, characterized by the decline of memory function. There are few studies that prove the relationship between Nrf2 disruption and AD; the neuroprotective effect is mainly attributed to GSK-3B in the regulation of the Nrf2 pathway [105].

#### 3.3.4. Nrf2 in Liver Injury 

In liver regeneration, *Nrf2*KO mice showed a significantly reduced capacity to initiate cell proliferation in 72 h after a two-thirds partial hepatectomy as an outcome of blunted Notch1 signaling; the proximal region of the Notch1 promoter contains an ARE sequence [106]. Nrf2 is additionally implicated in oxidative stress induced by alcohol consumption, upregulating antioxidant defense genes and downregulating the genes involved in lipogenesis. Nfr2-deficient mouse hepatocytes attenuate phosphoinositide 3-kinase (PI3K) protein kinase b (AKT) signaling affecting the phosphorylation of AKT, GSK3, and PPP, and the cell-proliferation efficiency established regarding the PI3K/AKT pathway [107,108].

In liver regeneration, *Nrf2*KO mice exhibited significantly reduced capacity to initiate cell proliferation in 72 h after a two-thirds partial hepatectomy as an outcome of blunted Notch1 signaling; the proximal region of the Notch1 promoter contains an ARE sequence [106]. Nrf2 is additionally implicated in oxidative stress induced by alcohol consumption, upregulating antioxidant defense genes, and downregulating the genes involved in lipogenesis. Nfr2-deficient mouse hepatocytes attenuate phosphoinositide 3-kinase (PI3K) protein kinase b (AKT) signaling, affecting the phosphorylation of AKT, GSK3, and PPP and the cell-proliferation efficiency established regarding the PI3K/AKT pathway [107,108].

## 4. Nfr2 Bioactivators

In pharmacology, the so-called “Nrf2 inducers” are Keap1 inhibitors that promote Nrf2 dissociation and nuclear translocation. Therefore, these molecules can be classified as electrophiles, protein-protein interaction inhibitors, and multitarget drugs [109]. Electrophilic compounds covalently modify the cysteine residues present in thiol-rich Keap1 protein by oxidation or alkylation [79,85]. Nrf2 inducers have the ability to react with sulfhydryl groups (-SH); they can modify Keap1, or oxidize one or more of its cysteine thiol groups; the most susceptible of these to the electrophilic reaction appear to be Cys-151, Cys-273, Cys-288, Cys-226, Cys-234, and Cys-613 [110,111]. Inducers are a mechanism to inhibit Nrf2 ubiquitination by Keap1 sequestration by means of the modification of cysteine residues, generating a weak interaction between Nrf2 Neh2 motifs (DLG and ETGE) and the Keap1 dimmer. In this regard, Nrf2 is newly synthesized because Keap1 is not regenerated at a sufficient rate; then Nrf2 escapes from Keap1 [112]. Certain evidence establishes that upstream kinases may phosphorylate specific threonine Nrf2 residues, disrupting Keap1/Nrf2 bonding. Therefore, this may aid nuclear translocation [113]. 

Understanding how Nrf2 could be transcriptionally activated has attracted attention to a wide variety of molecules that may exert some effect on this. Several natural compounds or phytochemicals have been identified as electrophilic Nrf2 inducers. These inducers are also denominated “Nrf2 bioactivators”, but only a few of these formed among the molecular mechanisms have been extensively studied. A number of Nrf2 inducers, mostly plant-derived compounds such as sulforaphane (SFN) from broccoli [114], curcumin from turmeric [115,116], and resveratrol from grapes [117], have proven to active the Nrf2 cytoprotective pathway. Special attention has been paid to the previously mentioned compounds due to their chemopreventive properties as described in numerous clinical trials. SFN has been one of the most studied among these [118].

One way to estimate Nrf2 induction is through the expression of NQO1, which is considered a cytoprotective enzyme responsible for maintaining cellular defenses. It is highly active in endothelial cells, epithelial cells, and lung tissues. NQO1 activity is used as a reference to prove the anticancer activity of phytochemicals. In this regard, the comparative biomarker is the “CD value”, which describes the concentration of any compound utilized to double NQO1 activity in murine hepatoma cells [119]. Considering that the lesser amount is that required to double the activity of NQO1, the most potent of these is the phytocompound SFN, which appears to be the most effective inducer, with a concentration of 0.2 μM, followed by and ragrapholides (1.43 μM), quercetin (2.5 μM), curcumin (2.7 μM), SM (3.6 μM), tamoxifen (5.9 μM), genistein (6.2 μM ), beta-carotene (7.2μM), lutein (17 μM), resveratrol (21 μM), indol-3-carbinol (50 μM), chlorophyll (250 μM), alpha-cryptoxanthin (1.8 mM), and zeaxanthin (2.2 mM) [120].

SFN is derived from the precursor glucoraphanin, which is contained inside of the vacuoles with the enzyme myrosinase in *Brassica* species. Then, SFN is formed when the plant cell ruptures and glucoraphanin and myrosinase come into contact. Broccoli is the crucifer with the higher content of glucoraphanin, around 75%, of all of the glucosinolates [121]. However, once SFN is produced, it begins to degrade quickly because it is less stable than its precursor; hence, the procedures of chewing and cutting not only trigger SFN synthesis, but its degradation as well [122]. SFN as a natural isothiocyanate produces Nrf2 activation by changing the covalent linking of the cysteine residues of Keap1 by direct reaction with the electrophilic isothiocyanate group. Growing evidence supports the efficacy of SFN as a potent Nrf2 activator in a large number of investigations related to cancer [77] in neurodegenerative disorders [123]. Curcumin is another potent bioactivator that modifies Cys-151 in Keap1. It has been shown to suppress tumor genes by activating Nrf2 [124] and, in addition, it possesses antioxidant and anti-inflammatory activity [116].

## 5. Silymarin and Flavolignans as Potent Nrf2 Inducers

### 5.1. Silymarin and Flavolignans Description

Silymarin (SM) is a mixture of flavolignans extracted from the seeds of species derived from *Silybum marianum*, commonly known as milk thistle or St. Mary’s thistle. Originally, it is a native species from the Mediterranean region, grows in its natural form in Southern Europe, the Middle East, Southern Russia, Northern Africa, and in America, and is cultivated in the North and South. Milk thistle belongs to the *Asteraceae* family, is a plant with a height of 0.9–1.8 m [125], the flowers are 4–8 cm in diameter, bright pink and magenta in color, each containing around 50–200 small individual tubular florets, which form groups of flowers as well. Its leaves reach up to 30 cm in width and 75 cm on length and are characterized by milky white veins with a smooth and hairy surface, thus its name of milk thistle [126] *Silybum marianum* has been used from many years ago in traditional medicine in Greece and Asia. In China, its use had been reported since at least 2000 years ago due to its properties in the treatment of liver disorders. Herbalists and physicians in ancient times described it as a nephro-, neuro-, hepato-, and cardioprotective because of its antioxidant, anti-inflammatory, and regenerative effects [105,127]. 

The chemical composition of SM comprises mainly compounds with 25 carbon atoms and a flavonoid combination of 65–80% of seven flavolignans; the most important of these include silybin, isosilybin, silychristin, isosilychristin, and silydianin. Silybin is the most abundant compound in around 50–70% in isoforms silybin A and silybin B, and then followed by the remaining isoforms: isosilybin A; isosilybin B; isosilybin C; isosilybin D; silychristin A, and silychristin B (Figure 3). Other flavonoids are present, such as quercetin, kaempferol, taxifolin, apigenin, etc. Proteins, sugars (arabinose, rhamnose, xylose, and glucose), tocopherol, sterols, fatty acids (linoleic, oleic, and palmytic) and alcohols are found in small quantities [125]. Therefore, silybin as the major constituent has been attributed to being responsible for the effects of SM, is transformed to 2,3-dehydrosilybin as oxidation product (Figure 4b). Since the year 1959, from the discovery of silybin, many studies have been conducted in order to identify its different biological activities, such as antioxidant, chemoprotective, anti-inflammatory, and others [128,129].

The silybin molecule can be described as small, with carbo- and heterocycles of two units, one of which is a flavonol group, taxifolin, and the other, a unit of coniferyl alcohol phenylpropanoid; both structures are linked by an oxirane ring [130]. The chemical structure of silybin was first described in 1968, but its complete position and configuration were not reported until 1975. The molecule is stable in acid solutions and under prolonged overheated conditions, the structure is chemically modified and unsterilized. It is highly soluble in polar aprotic solvents such as dimethyl sulfoxide (DMSO), acetone, tetrahydrofuran (THF), and *N*,*N*-dimethylformanide (DMF), less soluble in ethanol or methanol, and insoluble in non-polar solvents such as chloroform and petroleum ether. Additionally, silybin can be easily oxidized into 2,3-dehydrosilybin by a couple of oxygen molecules. It has five hydroxyl groups on its skeleton, but only 5-OH, 7-OH, and 20-OH have phenolic groups. The first of these possesses, in conjugation with the phenolic ring, a strapping bonding between hydrogen and the oxo group, the latter conferring an electron pair donor on the hydrogen bond with the 5-OH group. The remaining two OH phenolic groups have very similar characteristics [130]. 

### 5.2. Silymarin and Flavolignans Pharmacokinetics

#### Bioavailability and Metabolism

Once it is orally ingested, SM flavolignans are poorly absorbed because their molecules are too large to be absorbed by simple diffusion. Some previous studies reported 0.073–0.95% of bioavailability in rats [131,132]. These flavolignans are absorbed in the gastrointestinal tract, reaching a t_max_ of 2–4 h, and interact along enterohepatic circulation with a half-life elimination of 6–8 h. The issue of bioavailability is affected by low water solubility, due to theirnon-ionizable and hydrophobic structure, affecting their ability to cross the lipid membranes of the small intestine [133]. Low water solubility at around 0.04 mg/mL has been reported [131]. In addition, SM flavolignans have poor miscibility with other lipids, thus resulting in a limited capacity to be absorbed in the rich-lipid outer membrane of the enterocytes in small intestine [134]. Their absorption also depends on several factors, such as the purity and concentration of the extract and the presence of other substances in the preparation, such as proteins, sugars, vitamins, and other polyphenols, processes that have their own physical and chemical properties and which could interfere in the absorption process. The previous study revealed a maximal serum concentration (C_max_) within a range of 0.18–0.62 μg/mL after the oral administration of 240 mg of silybin [135]. The dose-escalation study run by Zhu et al. [136] evaluated the pharmacokinetics of the six major SM flavolignans in healthy volunteers. These authors found that, after receiving single oral doses of 175, 350, and 525 mg of standardized milk-thistle extract (Legalon^®^) for 28 days, the flavolignans were rapidly absorbed and eliminated. SM flavolignan concentrations were linear and dose-proportional as assessed by mean C_max_ and AUC_0–24_. In order of concentration, the mean ± SD C_max_ major compound found was silybin A (106.9 ± 49.2, 200.5 ± 98, and 299.3 ± 101.7 ng/mL) followed by silybin B (30.5 ± 16.3, 74.5 ± 45.7, and 121.8 ± 52.2 ng/dL), isosilybin A (6.1 ± 2.9, 18.2 ± 13.5, and 24.7 ± 11.8 ng/dL), isolilbyn B (22.0 ± 10.7, 46.4 ± 31, and 75.8 ± 32.3 ng/dl), and silychristin was detected at doses of 350 and 525 mg (4.6 ± 1.1 and 8.5 ± 3.4 ng/dL), as was silydiadin (6.5 ± 3.8 and 5.1 ± 2.7 ng/dL). Median t_max_ values fell within a range of 1.0–1.5 h at the three different dosages for all flavolingnans. There were no significant differences in apparent clearance among the groups. Certain investigations have established that SM flavolignan availability depends on the manner of administration [137].

In order to improve SM flavolignan bioavailability, many attempts have been made to increase their solubility through the creation of emulsions, suspensions, or micelle-mix solutions. Sodium cholate phospholipid silybin-mixed micelles proved to be an effective way to deliver silybin and improve bioavailability [133]. The design of silybinnano-emulsions using oil, surfactants, and co-surfactants (sefsol-218/Tween 80/ethanol) in oral administration was more capable of improving the SM hepatoprotective effect than SM alone [138].After the oral administration of 20 mg/kg nanosuspension (0.2% lecithin and 0.1% poloxamer 188) in dogs, an increase was observed of C_max_ (2.73 ± 0.30 μg/mL vs. 1.53 ± 0.22) [139].

Conjugation comprises the main biotransformation route identified for SM and its metabolites; nevertheless, silybin is biotransformed by both phase-I and phase-II reactions in liver cells [140]. About 80% of silybin metabolites are excreted as glucuronides and sulfate conjugates by the UDP-glucuronosyltransferases (UGT) and Sulfotransferases (SULT). In-vitro studies suggested UGT1A1 as mainly responsible for the glucuronidations of silybin A and B [141,142]. In this regard, it can be observed that silybin mono-, di-, and sulphoglucuronides, all of these formed in phase II, and 31 metabolites have been identified [131]. Around 3-8% of the silybin consumed is eliminated in an unchanged form, whereas concentration in bile is much higher than in serum (60–100-fold): possibly 20–40% returns to the system and the remainder is excreted in feces [125]. Recently, the study of the *UGT1A1*28* polymorphism reported that the latter has been associated with a reduction of glucuronidation [143]; consequently, this could affect the pharmacokinetics of drugs [144,145]. Nonetheless, the presence of *UGT1A1*28* in patients with liver disease does not appear to significantly affect the pharmacokinetics of silybin A and silybin B due to an apparent compensatory increase of sulfate conjugation. However, due to its large intervariability, it might be difficult to understand the possible beneficial effects of SM for patients with liver diseases [146].

Silybin biotransformation has been associated with CYP450 2C8 with the resulting major form *O*-demethylatedsilybin and minor metabolites mono- and di-hydroxysilybin [147,148]. On the other hand, it has been reported that SM and its metabolites may inhibit several cytochromes including CYP540 3A4, 2C9, and 2C8 [107,149]. Nonetheless, recently it was recently demonstrated that SM does not interfere in their activities [150].

### 5.3. Nrf2 Activated by Silymarin and Flavolignans: Promising Therapeutic Model 

For many years, the antioxidant activity of SM has been frequently mentioned in the scientific literature. It appears that SM can work in diverse ways to reduce or prevent oxidative stress, acting directly on the scavenging of free radicals, blocking specific enzyme producers of free radicals, maintaining the integrity of the electron transport chain, contributing to optimal redox cell status, enhancing enzymes of the endogenous antioxidant systems, inducing non-enzymatic antioxidant defenses such as glutathione or transcription factors (Nrf2 and NF-κB), promoting signaling pathways of the redox response and, finally, activating the specific genes responsible for producing protector proteins such as sitruins, thioredoxins, and HSP [12].

As mentioned previously, Nrf2 plays a crucial role in the defense response in cooperation with other transcription factors to mediate the insults of oxidative stress. Recently, several molecules such as polyphenols, which have been identified with antioxidant activity, have proven to participate in the activation of the Nrf2/Keap1/ARE pathway, regulating the expression of cytoprotective antioxidant enzymes such as NADPH and NQO1, this explained by means of the molecular basis supporting the described activity. In current investigations, SM and its flavolignans have shown to activate Nrf2 (Table 1). The research of Roubalová et al. [151] found that, after 6 h of exposure of the main *Silybum marianum* flavolingnans and their 2,3-dehydroderivatives, 25 μM and 50 μM of 2,3-dehydrosolydianin significantly increased the activation of Nrf2 and the expression of the *Nqo1* gene, 1.6- and 2.3-fold, respectively, as well as of the target genes Hem oxygenase1 (*Hmox*-*1*), gammaglutamyl-cystein ligase modifier subunit (*Gclm*), and gamma glutamyl-cystein ligase catalytic subunit (*Gclc*) by 2.2-fold, 1.5-fold, and 1–3-fold, respectively, but only at a concentration of 50 μM against the control in murine Hepatoma Hepa1c1c7 cells. In addition, the concentration of NOQ1 and GCLM proteins also increased at 50 μM, whereas the levels of HMOX1 and GCLC did not change significantly. Meanwhile, the remaining compounds did not exert a significant effect on NQO1 activity. The results suggest that 2,3-dehydrosilydianin activates Nrf2 and the expression of target cytoprotective genes. In conclusion, some of the effects of SM can be attributed to minor flavolignans or other components such as quercetin and taxifolin, which have been recognized as Nrf2 activators. Thus, SM sulfate compounds employed at doses of 50 Μm for 48 h in Hepa1c1c7 cells caused 1.2-fold increases in the activity of NQO1 by silybin A 20-*O*-sulfate, of 0.9-fold by silybin B 20-*O*-sulfate, of 1.3-fold by 2,3-dehydrosilybin-20-*O*-sulfate, of 1.2-fold by 2,3-dehydrosilybin-7,20-di-*O*-sulfate, of 1.3-fold by silychristin-19-*O*-sulfate, and of 1.4-fold by 2,3-dehydrosilychristin-19-*O*-sulfate compared with the control. In addition, the sulfated 2,3-dehydroderivatives appeared to be more active in radical scavenging with Ferric (FRAP) and Folin–Ciocalteu reagent (FCR) reducing activity compared with the original molecules [152].

In a model of oleic acid-treated HepG2 cells employed as an in-vitro model of steatosis, oxidative stress, and insulin resistance, treatment with silibinin at 5, 20, 50, and 100 μM for 24 h decreased the cellular levels of triglycerides and nitric oxide, induced the activation of the death domain-like apoptosis regulator (CFLAR-JNK) pathway, of the target gene sterol regulatory element-binding protein1C (SREBP-1C), peroxisome proliferator activated receptor-α (PPAR-α), the patatin-like phospholipase domain containing three (PNPLA3), which are related to lipid metabolism, oxidative stress genes *Nrf2*, *CYP2E1*, and *CYP4A*, the changes in glucose uptake with the upregulation of proteins phosphatidylinositol 3 kinase (PI3K), and phosphorylated serine-threonine protein kinase (pAKT). These results suggest the possible ameliorating effect on non-alcoholic steatohepatitis (NASH) [155]. The investigations conducted by Ouet et al. [156] on male C57BL/6 mice revealed, after 8 weeks of a methionine-choline-deficient diet, the induction of NASH. The oral ingestion of silybin significantly inhibited the gene expression associated with lipid metabolism, inflammation-related gene expression, and the NF-κB signaling pathway. Moreover, silybin treatment activated the Nrf2 pathway, favoring the upregulation of target antioxidant genes in the silybin-treated group, modulating oxidative stress. 

Damage produced by carbon tetrachloride (CCl_4_) is commonly utilized as a liver injury model due to modifications in the activation of the apoptotic and fibrotic pathways and rising oxidative stress. These processes are mediated by transforming growth factor beta (TGF-β), inducing the expression of oncogenes, cytokines, and the activation of small mother against decapenaplegic (Smad). Stimulation of thecyclooxygenase-2 (COX-2) and vascular endothelial growth factor (VEGF) pathways is related to the activation of PI-3/AKT focal adhesion kinase-phosphatidylinositol-3-kinase, the signal transducer and activator of transcription-3 (STAT-3) and mitogen-activated protein kinase (MAPK). In this regard, oral administration of SM in combination with vitamin E and/or curcumin in rats reduced the expression of fibrogenic and apoptotic factors and increased the expression of Nrf2 and cytoprotective antioxidant defenses. This suggests the combination of SM with vitamin E and/or curcumin can be a good option for the treatment of liver injury induced by toxic substances, in that the antiapoptotic effect could be potentiated [160]. Furthermore, the pretreatment of a combination of *S*-adenosylmethionine (SAM) 30 and 2000 ng/mL and silybin 298 ng/mL in canine hepatocytes increased antioxidant enzyme-reduced glutathione compared to the control, and the treatment attenuated cytokine-induced prostaglandin E2, Interleukin1β (ILβ), and the primary chemotactic protein-1 (MCP-1) produced by the activation of NF-κβ [163].

In KYSE70 cells, SM (100 μM), artichoke (1 mg/mL), and propolis (1%) were potent activators of UGT1A1 enzyme activity all the way through the Aryl hydrocarbon receptor (AhR) and the Nrf2 pathway, significantly reducing the oxidative stress that resulted from the hydroperoxide levels induced by tertiary butylhydroquinone (tBHQ) (100 μM). UGT´s are responsible for glucuronidation in phase II of metabolic biotransformation and comprise the major route for xenobiotics in the organism. Therefore, the presence of silymarin, artichole, and propolis could affect the metabolism of other drugs when these are metabolized by glucuronidation [153].

In a cardiorenal-injury model in rats, the combination of the ethanolic extract of *Vitis vinifera* (500 mg/kg) and the SM extract (600 mg/kg) significantly increased the expression levels of Nrf2 linked to the redox-sensitive pathway. Further, cardiac and renal biochemical indicators, such as creatinine, urea, BUN, and the lipid profile, improved significantly, as well as histopathological alterations [154]. The experimental model of Indomethacin-induced gastric injury in albino rats had proven that oral pre-treatment with 50 mg/kg of SM inhibits the synthesis of lipid peroxides, promotes the upregulation of Nrf2, and the enhancement of the activity of GPx and SOD enzymes, increasing antioxidant and cytoprotective defense, thus preventing gastric oxidative stress. Gastric inflammation improved by inhibiting tumor necrosis factor alpha (TNF-α), interleukin6 (IL-6), and myeloperoxidase activity, in conjunction with the expression of NF-Κβ, as well as the reduction of caspase-3. The histological analysis revealed mild sub-mucosal edema and inflammatory edema infiltrates and minimal alterations in epithelium surface in the group pretreated with SM. These results suggest that SM exerts a gastro-protective effect [157]. Otherwise, the co-administration of silybinin at 75 mg/kg/day in a rat experimental model with sodium arseniate (NaAsO_2_) (5 mg/kg/day) during 4 weeks induced kidney toxicity that resulted in a decrease of lipid peroxidation (lipid hydroperoxides, protein carbonyls, and TBARS), NADPH oxidase, iNOS, and NF-kB, also inhibiting caspase-3-mediated tubular apoptosis. Silibinin upregulated the Nrf2 pathway in renal tissue, improving the levels of enzymatic (SOD, CAT, GPx, and GST) and non-enzymatic (GSH, TSH, and vitamins C and E) antioxidants; additionally, silybinin treatment reduced serum and urine markers of nephrotoxicity (urea, ureic acid, and creatinine clearance), reduced tubular necrosis, degeneration, and dilatation, as well as thickened basement membrane and desquamation. Glomeruli atrophy and diffuse hemorrhage were significantly reduced in the treated group. The study established the potential nephro-protective effect of silibinin against renal arsenic toxicity [162].

Paraquat is used as a potent biohazardous herbicide that produces superoxides when it is reduced. In consequence, it reacts with the membrane of unsaturated fatty acids, causing severe human fatality. Paraquat is being employed as a model of lung injury; the previously mentioned study points out that SM can be a potential element for lung-injury therapy. According to the results of the investigation conducted by Zhao [161], treatment with SM at 200 mg/kg for 3 days improved oxidative stress by reducing MDA and increasing the activity of SOD, Cat, and GPx in lung tissue and serum in rats treated with Paraquat (30 mg/kg) to induce lung injury. These results were related to the upregulation of Nrf2, HO-1, and NQO1 in male Sprague-Dawley rats. At the same time, SM ameliorated histopathological changes in lung and in proinflammatory mediators, suppressed myeloperoxidase activity, Nitric oxide (NO)/inducible Nitric oxide synthase (iNOS) expression, reduced inflammatory cell infiltration, and the lung Wet weight/Dry weight (W/D) ratio. These mechanisms were associated with the Nrf2 pathway. Previously, the protective effect was investigated of SM against Paraquat-induced oxidative stress on the human A549 adenocarcinoma cell line. SM exerted an effective cytoprotective effect by reducing cell toxicity on inducing antioxidant genes *Nrf2*, *NQO1*, and *HO-1* after only 3 h of exposure. Hence, the evidence supports SM in aiding in Paraquat intoxication [164].

In neurological models, the use of SM (2, 4, and 10 μM for 4 h) had shown to be effective as well. In an Alzheimer’s-disease model of HT-22 hippocampal cells with Aβ_25–35_-induced OS injury at 2 μM for 24 h, isosilybin induced the expression of Nrf2 promoted by translocation into the nucleus, thus stimulating the activity of an antioxidant response element (ARE), activating the Nrf2/ARE signaling pathway, and regulating the expression of HO-1, GST, and aldo-keto reductases 1C1 and 1C2 (AKR1C2), plus significantly inhibiting ROS production, the release of malonaldehyde (MDA) and lactate dehydrogenase (LDH), alleviating the increase of oxidative stress in this model [159]. The acrylamide-induced neurotoxicity model in PC12 cells revealed that SM at different concentrations (12, 24, 48, 96, and 192 μg/mL for 3 h) induced protection in a dose-dependent manner against damage caused by the exposure of 5 mM acrylamide for 24 h. SM facilitated Nrf2 translocation from cytosol into the nucleus, the mRNA and protein expression levels of Nrf2, as well as cytoprotective genes *GPx*, *GCLC*, and *GCLM*. These results correlated with the increased levels of intracellular GSH, reduced levels of ROS, and MDA. In conclusion, SM can exert neuroprotection against acrylamide-induced damage [158]. 

In the experiment conducted by Choi et al. [165], the effect was investigated of SM and (−)-Epigallocatechin 3-*O*-gallate (EGCG) supplementation on gluconeogenesis and lactate production during exercise in a rat model. After 4 weeks of exercise training on a motor-driven treadmill (60 min/5 days/week) at a speed of 8 m/min, a marked reduction was observed of lactate and triglyceride levels in SM+Ex and EGCG+Ex groups compared with control groups, although glucose level, body weight, and liver weight did not change. Insulin signaling was suppressed along the exercise training in liver, increasing glugoneogenesis from lactate in active muscles due to the reduction in Akt phosphorylation involved in the insulin signaling pathway. SM+Ex and EGCG+Ex exhibited an upregulation of phosphoenol pyruvate carboxykinase (PEPCK) and peroxisome proliferator-activated receptor gamma (PPARγ), which are both involved in β-oxidation and gluconeogenesis pathways compared with the exercise alone group. In addition to the latter, 5-AMP activated protein kinase (AMPK) and energy cell sensor and Akt phosphorylation was decreased in the gastronemius and soleus muscles of the supplemented groups. Also, pyruvate dehydrogenase kinase 4 (PDK4) expression in both types of muscle was higher in SM- and EGCG-administered groups. These results suggest that supplementation with SM during exercise modulates the metabolism of glucose, lipids, and lactate, improving endurance. Despite that Nrf2 was not determined in this investigation, the results supported the beneficial effects of SM and exercise in the treatment of the metabolic syndrome.

As previously mentioned, although several other mechanisms of action are suggested for the action of SM [12], the effects on Nrf2 activity appear to be crucial. Recent works describe the action of compounds on this system by direct interaction or by the disruption of proteins linked to Nrf2 activity [166,167], with Keap1 as the main target [168],due to that, under basal conditions, the Nrf2 protein is constitutively trapped by Keap1 and retained in the cytoplasm for ubiquitin conjugation and subsequent proteasome degradation. In this regard, 178 phytochemicals proven to possess good antioxidant properties were selected and tested in silico by Li et al., suggesting that phenylethanoid glycosides, tocopherols, flavones, flavanols, anthocyanins, and flavonols have entertained high affinity with Keap1 and have potential as inhibitors of the inactivator action on Nrf2 [169]. In a complementary manner, itis noteworthy that several research workgroups have developed and tested compounds to disrupt Keap1:Nrf2 interaction [168,170,171]. The main aim of these approaches is the design of potential drugs for numerous pathologies, but also for deciphering molecular disease mechanisms [170]. Interestingly, the binding site shares several residues conserved for ligands with various aromatic rings (often 3 or 4, as several bioactive compounds from SM), albeit that tested compounds exhibit notable differences, such as those identified from a peptide series that demonstrated improved binding affinity in fluorescence polarization, differential scanning fluorimetry, and isothermal titration calorimetry assays [168]. In the meanwhile, other potent inhibitors were synthesized strategically by means of the exploration and optimization of protein-ligand interactions in three energetic “hot-spots” identified by fragment screening [170]. Here, with solely illustrative purposes, SM compounds were tested on Keap1 by conventional docking procedures. In brief, the structures of ligands were drawn using ChemBioDraw Ultra ver. 12.0. Their geometry was pre-optimized by using Hyperchem (version 6.0; Hypercube, Monterey, CA, USA; [http://www.hyper.com]) at the level of molecular mechanics (AM1 basis set). Therefore, the minimal energy structure for each ligand was fully optimized at the B3LYP/6-31G** level by using Gaussian 09 software [172]; then, they were tested on the 3D crystal structure of Keap1 (PDB ID: 6QMC). For the latter, a grid-box of 70 × 70 × 70 Å was centered on the main crevice of Keap1. We utilized a Lamarckian genetic algorithm to perform the search with an initial population of 100 random individuals, 1.0 × 10^7^ interactions were run with the AutoDock ver. 4.2.6 program, and the results were analyzed by using Autodock Tools and VMD ver. 1.9.2 software and determining binding poses and affinity as elsewhere. Although additional in-silico procedures are required in order to evaluate the affinity and consequences of binding, these assays reveal the feasibility of these compounds for reaching the same binding site as that of the recently reported ligands (Figure 5). Main contacts are by means of electrostatic and Vander Waals interactions, as well as that some hydrogen bonds occur with sites clearly linked to the Keap1-Nrf2 interphase (the detailed analysis of interactions is required, but due to that, in that it falls beyond the purpose of the current contribution, it is not presented herein). Consequently, it is expected that several compounds from SM are able to perform putative disruption of the Keap1-Nrf2 complexes, leading to the higher activity of Nrf2.

## 6. Conclusions and Perspectives

Transcription factor Nrf2 is one of the most interesting topics to investigate in molecular science, this attributable to its action as a master regulator of the antioxidant and cytoprotector response. The sensitive redox signaling pathway Nrf2/Keap1/ARE is transcendental for the regulation of physiological and pathological conditions. Even more so, the mechanisms of regulation of this complex system are not fully understood, but it is now well-known that the Nrf2 pathway is implicated in several diseases, such as diabetes, neurological, cardiovascular, liver, kidney injury, and cancer [75]. Although there are existing paradoxes in cancer because growing evidence indicates that Nrf2/Keap1/ARE participate in carcinogenesis, it has been demonstrated that Nrf2 could prevent the process at early stages [76]. 

The Nrf2/Keap1/ARE system comprises a very promising pharmacological target for managing the pathological process of numerous diseases characterized by inflammation and oxidative stress [173]. Emergent research is centered on Nrf2 activators, and few of these are under clinical investigation. Bioactivators are compounds deriving from plants or foods that have been proven to activate Nrf2. Due to its electrophilic nature, SFN is the most studied among inducers [114]. However, there are other interesting compounds, such as curcumin, SM, and resveratrol, which demonstrate potential in the induction of Nrf2. SM has a long history in therapy for liver damage and including its the protector and antioxidant properties [12]. In addition to its low bioavailability, current reports indicate that it possesses the potential to activate Nrf2/Keap1/ARE. Further researches with in silico, in vitro, and in vivo models are necessary to elucidate the possible manner in which SM and/or its flavolignans are able to modulate the Nrf2/Keap1/ARE system. The latter is a possibility for novel therapeutic approaches in the management of diseases in the present or near future.

## Figures and Tables

**Figure 1 biomedicines-08-00122-f001:**
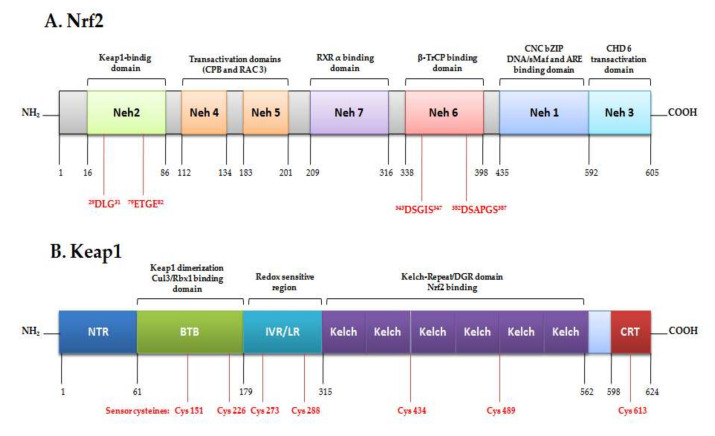
Original figure representing structures of nuclear factor erythroid 2 (NFE2)-related factor 2 (Nrf2) and Kelch-like ECH-associated protein (Keap1) domains and their interactions.

**Figure 2 biomedicines-08-00122-f002:**
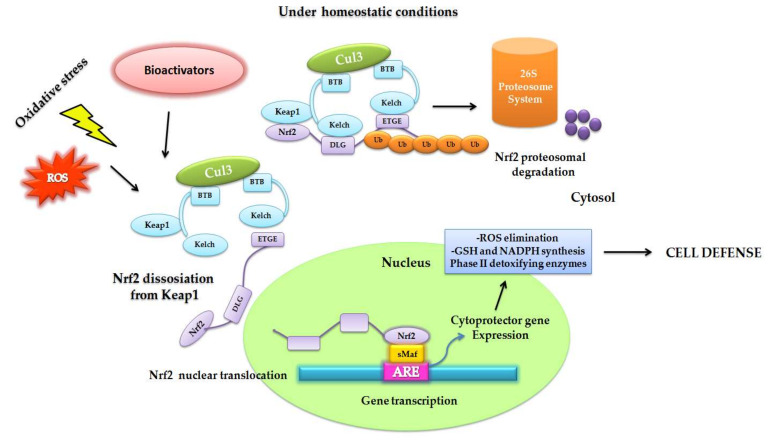
Original figure representing how the rising oxidative stress and accumulation of ROS or the presence of bioactivators molecules triggers the Nrf2-Keap1 signaling pathway. Under homeostatic conditions, Nfr2 is bound to Keap1 through the DLG and EGTE motifs in the Neh2 domain of Nrf2 by means of the ubiquitin ligase complex Cullin (Cul)3-RING-box protein (Rbx)1 (Cul3). Nrf2 is ubiquitinized due to its rapid proteosomal degradation. The electrophilic molecules produce the oxidation of the cysteine residues in Keap1, favoring its conformational change; consequently, ubiquitination is impaired and Nrf2 dissociates itself from the inhibitor complex. Nrf2 accumulates and is maintained transcriptionally active as it proceeds into the nucleus. It futher heterodimerizes with small musculo-aponeurotic fibrosarcoma proteins (sMaf), bonding in a specific DNA sequence denominated the antioxidant response element (ARE), inducing the expression of cytoprotector genes to increase cellular defense: the elimination of ROS; the gluthathione (GSH), and NADPH synthesis, and the expression of phase-II detoxifyng enzymes.

**Figure 3 biomedicines-08-00122-f003:**
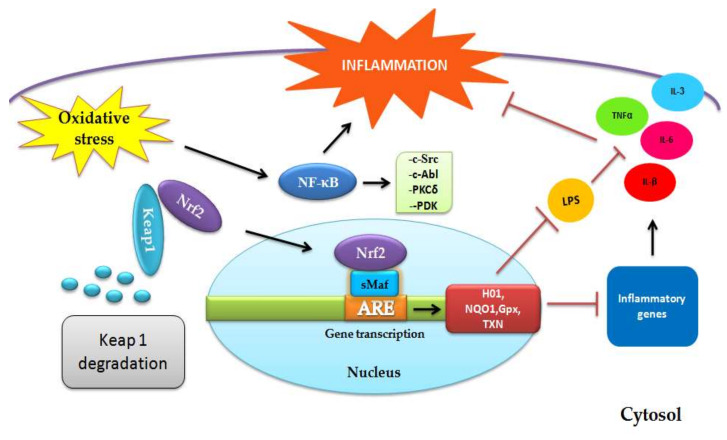
Original figure representing the Nrf2 role on inflammatory process.The activation of Nrf2/ARE pathway promotes the transcrpition of protector genes heme-oxygenase1 (HO-1), NAD(P)H-quinone oxidoreductase-1 (NQO1), glutathione peroxidase (GPx), and thioredoxin 1 (TXN) inhibiting expresión of inflammatory genes and the produción of cytokines IL-6, IL-3, IL-β, and TNFα, reducing inflammation. Moreover, the albation of Nrf2 induces high levels of ROS promoting inflammatory pathway via the nonreceptor proto-oncogene tyrosine-protein kinase c-Src, Abelson murine leukemia viral oncogene homolog 1 (c-Abl), protein kinase C (PKC)δ, and protein kinase D (PKD). Arrows in black mean the signaling pathway is triggered. Arrows in red mean the pathway is blocked.

**Figure 4 biomedicines-08-00122-f004:**
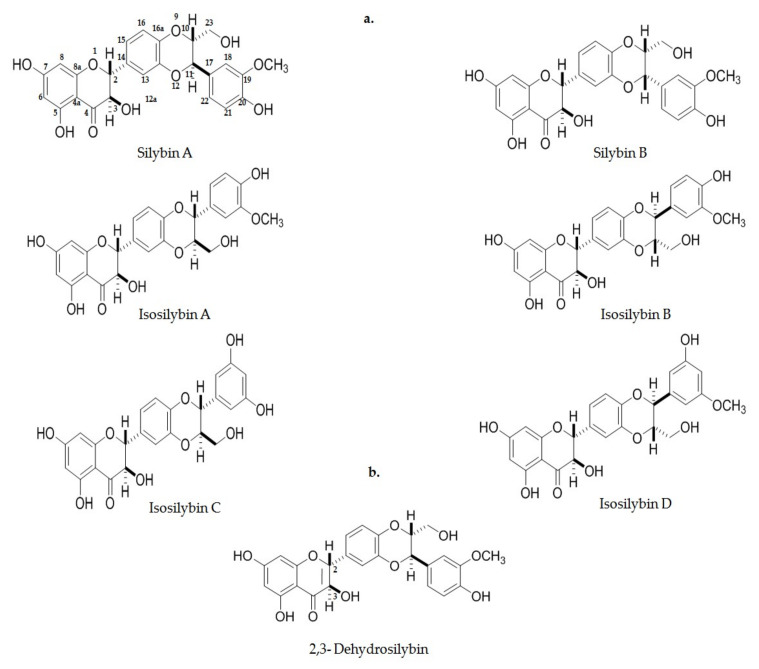
Chemical structure of the main compounds from silymarin. (**a**) Representation of molecules silybin A; silybin B; isosilybin A; isosilybin C and isosilybin D. (**b**) Oxidized form 2,3-Dehydrosilibyn structure.

**Figure 5 biomedicines-08-00122-f005:**
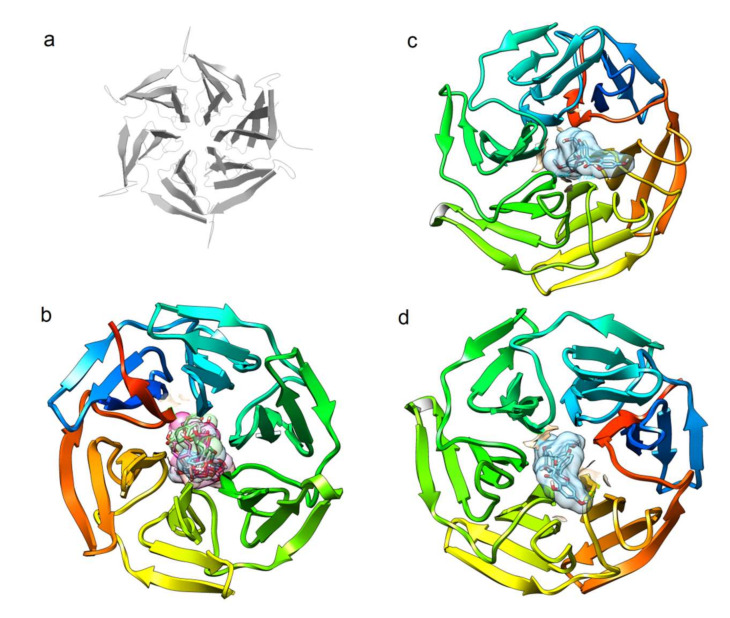
Theoretical feasibility of silymarin compounds to bind the Keap1. (**a**) Cartoon representation of the Kelch-domain crystallized with a nonamer Nrf2-derived peptide bound (PDB ID 6QMC). The peptide was removed for clear exposure of the putative area for contacting Nrf2. (**b**) The binding site of Keap1 for six flavolignans from silymarin (silybin A, silybinB, isosilybin A, isosilychristin, taxifolin, and silydianin), for silybin A (**c**) and for silybin B (**d**).

**Table 1 biomedicines-08-00122-t001:** Studies regarded to the activation of the Nrf2 signaling pathway in the presence of silymarin or its flavolignans.

Model	Protocol	Results	Reference
Human esophangeal squamous cell carcinoma (KYSE70) cells in OS model induced by tBHQ	-tBHQ (100 μM) -SM (100 μM) -Artichoke (1 mg/mL) -Propolis (1%) 48 h of exposition	↑ UGT1A1 enzyme activity by the AhR and Nrf2 pathway ↑ Induction of UGT1A7 with propolis, artichoke and SM (7.3, 5 and 4.5-fold respectively) ↑ UGT catalytic activity with porpolis, artichoke and SM treatment (21%, 29% and 20% respectively) in 90 min in the presence of substrate ↓ hydrogen peroxide were lower in tBHQ cells treated with phytochemicals	Kalthoff, et al. [153]
Rat cardiorenal injury model	Male Wistar albino rats -Doxorubicin via rat tail vein (3 and 2 mg/kg) to induce twice for 2 weeks to induce nephropathy -SM (600 mg/kg) for 4 weeks -Aqueous *Vitis vinifera* extract (400 mg/kg) for 4 weeks -Ethanolic *Vitis vinifera* extract (500 mg/kg) for 4 weeks -SM+EVVE (600 mg and 500 mg/kg) for 4 weeks	All the following results were significantly higher in SM+EVVE group: ↑ Nrf2 expression ↓ biochemical markers: BUN, urea, lipid profile ↓ hs-CRP and serum lipid profile ↓ histopathological alterations: improve cytomorphologic structure and restoration of myocardial architecture	Abdelsalam, et al. [154]
Hepa1c1c7 cells	Sulfate compounds from SM at 50 μM for 48 h exposition.	↑ activity of NQO1 compared to control: silybin A 20-*O*-sulfate1.2-fold, silybin B 20-*O*-sulfate0.9-fold, 2,3-dehydrosilybin-20-*O*-sulfate 1.3-fold, 2,3-dehydrosilybin-7,20-di-*O*-sulfate 1.2 fold, silychristin-19-*O*-sulfate1.3-fold. 2,3-dehydrosilychristin-19-*O*-sulfate1.4- fold	Valentová, et al. [152]
Oleic acid-treated HepG2 cell as in vitro model of steatosis, OS and IR	Silibinin at 5, 20, 50 and 100 μM for 24 h exposition	**↓** lipid metabolism genes: *SREBP-1C, PPAR α, PNPLA3* **↓** intracellular levels of TG and NO **↑** oxidative stress response genes:Nrf2, CYP2E1 and CYP4A ↑ glucose uptake: proteins PI3K and pAKT	Liu, et al. [155]
Male C57BL/6 mice a NASH model	-DL-methionine (3 g/kg) and choline bitartrate (2 g/kg) diet to induce NASH -orally ingestion of silybin at a dose of 105 mg/kg/day for 8 weeks	↓ Weight loss ↓ ALT and AST activities ↓ lipid metabolism gene expression ↓ NF-κB signaling pathway **↑** Nrf2 pathway: expression levels of GCLM, GCLC, NQO1, HOX1, GSTM1 **↑** levels of GPx and SOD in hepatic tissue **↓** histological disorders: hepatic steatosis, hepatocellular ballooning with inflammatory cell infiltration and liver fibrosis	Ou, et al. [156]
Rat gastric ulcer model	-5 days pre-treatment with ranitidine at 25 mg/kg orally (positive control) -5 days pre-treatment with SM at 50 mg/kg orally -Indometacin-induced gastric injury (25 mg/kg) in the last day	Prevent gastric OS by: ↑ Nrf2 expression, ↑ SOD and GPx activity ↓ gastric inflammation: TNF- α, IL-6 and myeloperoxidase activity, ↑ NF-κB expression Histological changes: protection observed as mild sub-mucosal edema and inflammatory cellular infiltrates and minimal alterations in epithelial surface	Arafa Kesnk, et al. [157]
Murine hepatoma Hepa1c1c7 cells	*S. marianum*flavolingnans and its 2,3- dehiydroderivates (25 μM) And 2,3-dehydrosilydianin (50 μM) 6 h exposition	↑ expression of Nrf2, *Gclm* and *Gclc*at 25 and 50 μM ↑ protein concentration of NOQ1 and GCLM at 50 μM HMOX1 and GCLC did not changed significantly	Roubalová, et al. [151]
PC12 cells Acrylamide-induced neurotoxicity model	Cells pre-treated with SM at 12, 24, 48, 96 or 192 μg/mL for 3 h, then cells were exposed to a 5 mM Acrylamide for 24 h	↑ mRNA and protein expression of Nrf2 in nuclear fractions ↑ translocation of Nrf2 from cytosol into the nucleus ↑ cytoprotective genes: *GPx, GCLC, GCLM* ↑ intracellular levels of GSH ↓ levels of ROS and MDA	Li, et al. [158]
HT-22 hipocampal cells AD model	Cells treated with: -Aβ_25-35_AT 2 μM for 24 h to induce toxicity in cells -Isosilybin exposition at 2, 4 and 10 Μm	↑ Nrf2/ARE signaling pathway: HO-1, AKR1C2 and GST ↓ ROS production ↓ cellular OS injury ↑ total antioxidant capacity in cells by ↓release MDA and LDH	Zhou, et al. [159]
Male Wistar rats CCl_4_ damage model	Hepatotoxicity induced with single dose of CCl_4_ (1 ml/Kg, IP) Orally administration of SM (200 mg/kg) alone or in combination with CA (60 mg/kg) and/or ME (20 mg (kg) for 21 days	↓ expression of fibrogenic and apoptotic factors ↓ Serum ALT activity ↓ liver caspase-3 activity ↓ hepatic CYP2E1 activity in SM, CA and ME treated group compared with SM alone ↓ oxidative DNA damage in liver: 8-OxodG levels	Al-Rasheed, et al. [160]
Male Sprague-Dawley rats Paraquat lung injury model	Paraquat exposition (30 mg/kg) to induce lung injury SM treatment (200 mg/kg) 3 days exposition	↓ MDA ↑ SOD, Cat, and GPx in lung tissue and serum. ↑ Nrf2, HO-1, and NQO1 expression ↓ Inflammatory cell infiltration and collagen deposition in the alveolar septum ↓ MPO activity and HYP content ↓ NO and iNOS ↓ Proinflammatory mediator levels: TNFα, IL-1β, IL-6 and the TGF-β1	Zhao, et al. [161]
Rat arsenic toxic model	Adult male Wistar albino rats treated with: -Arsenic (5 mg/kg) for 4 weeks silibinin at 75 mg/kg/day for 4 weeks	↓ lipid peroxidation, NADPH Oxidase, iNOS, NF-kB and TNFα ↓ lipid hydroperoxides, protein carbonyls and TBARS ↑ mRNA expression of Nrf2 and NADPH in renal tissue ↑ activities of enzymatic antioxidants: SOD, CA, GPx, and GST ↑ Non-enzymatic antioxidants: GSH, TSH, Vitamin C and E in kidney tissue	Prabu, et al. [162]

BHQ: tertiary butylhidroquinone; SM: Silymarin; UGT1A1: UDP-glucuronosil transferase 1A1; AhR: aryl hydrocarbon receptor; EVVE: ethanolic *Vitis vinifera* extract*;* hs-CRP*:* high sensitivity C-reactive protein; NQO1: NADPH quinoneoxido reductase 1; BUN: blood ureic nitrogen; IR: insulin resistance; *Gclm:* gamma glutamine cysteine ligase modifier subunit; *Gclc:* gamma glutamine cysteine ligase catalytic subunit; SREBP-1C: sterol regulatory element-binding protein-1C; PPAR α: peroxisome proliferator activated receptor-α; PNPLA3: patatin-like phospholipase domain containing 3; AKR1C12 aldoketo reductases 1C1 and 1C2; CYP2E1: cytochrome P4502E1; CYP4A: cytochrome P450A4; PI3K: phosphatidyl inositol 3 kinase; pAKT: phosphorylated serine-threonine protein kinase; AST: aspartate aminotransferase; ALT: alanine aminotransferase; NASH: non-alcoholic steatohepatitis; NF-κB: nuclear factor-κB; CCl_4_:carbon tetraclrorhidre; IP: intraperitoneal; CA: chlorogenic acid; ME: melatonin; TNF-α; tumer necrosis factor-α; IL-6: interleukin-6; IL-1β: interleukin-1β; TGF-β1: transforming grow factor-β1; AD: Alzheimer’s disease; MDA: malonaldehyde; LDH: lactate dehydrogenase; SOD: superoxide dismutase; Cat: catalase; GPx: glutathione peroxidase; HO-1: hemoxygenase-1; ROS: reactive oxygen species; GSH: Glutathione reduced form; TSH: total sulfhydryl groups; TBARS: thiobartituric acid reactive substances.

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
