# Peer review of "Flavolignans from Silymarin as Nrf2 Bioactivators and Their Therapeutic Applications"

_biomedicines, 2020, doi:10.3390/biomedicines8050122_

Round 1

Reviewer 1 Report

In this manuscript, the authors made an exhaustive and comprehensive review of the effects of flavolignans, from Sylimarin, as modulators of the Nrf2 signaling pathway. The role of this pathway in several diseases has been described in several research articles, thus this review compiles relevant information concerning the Nrf2 signaling, and its involvement in inflammation and several diseases. The interaction of sylamarin flavolignans with Nrf2 pathways is well described. This topic is very relevant, the descriptions are mostly scientifically correct, although the manuscript needs a revision prior to its publications. Along the manuscript there are several, imprecisions, misspell, grammar errors and scientific issues to be correct. I will attach the pdf of the manuscript in which are highlighted many expressions, units, verbs and sentences that need corrections.

Other points that require revision are be listed below, following the manuscript order.

Abstract: Please correct the scientific name of milk thistle, the correct is Silybum marianum (not Sylibium marianum).

Authors mention (1st line of abstract – line 33) that Silymarin (SM) is a mixture of flavonoids, and in line 36, of flavolignans. There should be a consistency in the terminology used along the document. Please correct.

Once an abbreviation is defined, authors should use the abbreviations and not a mix between the full word/expression and the abbreviation. Please correct.

Introduction:

Line 51: remove the capital letter in “Reactive” (“…of Reactive oxygen species (ROS)…”), or add capital letter in the other 2 words. A consistency in the use of capital letters, when refereeing to chemicals, enzymes and others, should be maintained along the document (see lines 55-56, and 95, for example).

Line 51: In the hydroxyl radical notation, the radical symbol should be placed near the oxygen (O) rather than near the hydrogen, please replace along the document.

Line 54, in my opinion the authors should not abbreviate oxidative stress (OS), as it is not used many times and many non-common abbreviations difficult the reading.

Line 86: “NFE2L2 gene “, NFE2L2 needs to be in italic. Please use italic in all genes notations (revise all document).

Line 86. Please correct the dalton (Da) abbreviation, replace “66 kD protein” by “66 kDa protein”,

Line 92. Does “NF-E2” refers to NFE2, described in line 67? If so, use the same terminology/abbreviation along the document. If not, please define.

In line 105, authors mention “Neh4 and Neh5 facilitate Nrf2 transcription”. This statement is not scientifically correct, as Nrf2 is a protein, and the term transcription defines a step of DNA based gene expression, it is used to genes. There is some confusion of terminology. Please revise, you may use the following references (doi:10.1042/BJ20061611).

Section 2, (line 82 to 108) would benefit of a figure or scheme showing the domain architecture of Nrf2 and its interactions.

Line 110, do the authors mean “Under low stress cellular conditions” or under homeostatic conditions? In homeostatic conditions there’s a basal level of oxidative stress. Please refine the terminology. The Nrf2 proteosomal degradation occurs under basal conditions.

Line 110, replace “little” by “low”

Line 118, please replace 26s by 26S.

Line 118 – 121, the sentence seems incomplete.

Line 122 and line 152, please be consistent in the use of small musculoaponeurotic fibrosarcoma proteins abbreviation (sMaf and Maf sre used). In Figure authors use sMaf. Please mention if Figure is original, or adapted, in case of adapted add the references.

Line 205. I would suggest using “reactive nitrogen species (RNS)” instead of ”nitrogen reactive species (NRS)”. This would be consistent with the expression “reactive oxygen species”.

Line 212: authors refer to the “deletion” of CuZnSOD1 enzyme. The term deletion is generally used to genes or to amino-acids. Please refine the sentence.

Line 217 – Please use the correct notation of the gene mSod2 (knock out KO, refers to a gene), and this work refers to a mice gene not a human gene, please see original Ref. Please correct the gene notations, rigorously. This one and in the next lines.

Line 256. A Figure to illustrate section 3.2. Nrf2 and Inflammation, would enrich the manuscript. The same also applies to the next sections. Please, add some figures.

Line 336. Please define NO in “Nrf2 NO mice”

Line 342: Please define: mTOR, mTORC1 mTORC2

Figure 2. Please correct the Figure notations (e.g. replace silybinB by silybin B, as in legend).

Line 468-469: The sentence is ambiguous, please clarify it. It is not clear if overheat destroys or not silybin.

Line 469, please define the solvents identified by DMSO, THF, DMF,

Line 470 – 476: It would be helpful if in Figure 2 silybin structure have the Carbons numbered. And authors could include an oxidized structure.

Line 481. Although the original article reports peak time as Tmax, the capitalized T is commonly used to temperature and not to time (which is t). Thus authors should define this quantity in the manuscript.

Line 481-484: Authors should mention that the described stomach absorption is not of free molecules but of molecules incorporated in micelles. Please clarify this point.

Please correct the English grammar concerning the verbal tenses, the number, and some spelling error. Also correct kilogram (kg not Kg), and other units highlighted in the manuscript.

Author Response

We are very pleased with the results of the evaluation conducted by the Scientific Committee of your renowned journal of our manuscript entitled “Flavolignans from Sylimarin as Nrf2 Bioactivators and Their Therapeutic Applications”, and have carried out the task of responding to the well-targeted and greatly appreciated observations provided by the reviewers.

In this document, please find, in yellow, the changes kindly suggested by the Reviewers, which now appear as changes in the new version of the manuscript. Many thanks.

We follow here with the corrections and suggestions that we carried out on the article, taking into account the reviewer commentary:

Reviewer1.

The answers of the reviewer are marked on the paper in color yellow

Comments and Suggestions for Authors

In this manuscript, the authors made an exhaustive and comprehensive review of the effects of flavolignans, from Sylimarin, as modulators of the Nrf2 signaling pathway. The role of this pathway in several diseases has been described in several research articles, thus this review compiles relevant information concerning the Nrf2 signaling, and its involvement in inflammation and several diseases. The interaction of silymarin flavolignans with Nrf2 pathways is well described. This topic is very relevant, the descriptions are mostly scientifically correct, although the manuscript needs a revision prior to its publications. Along the manuscript there are several, imprecisions, misspell, grammar errors and scientific issues to be correct. I will attach the pdf of the manuscript in which are highlighted many expressions, units, verbs and sentences that need corrections.

Other points that require revision are be listed below, following the manuscript order.

Abstract: Please correct the scientific name of milk thistle, the correct is Silybum marianum (not Sylibium marianum).

Answer. The mistake was corrected, thank you.

Authors mention (1st line of abstract – line 33) that Silymarin (SM) is a mixture of flavonoids, and in line 36, of flavolignans. There should be a consistency in the terminology used along the document. Please correct.

Answer. The terminology was corrected to flavolignans in the whole document.

Once an abbreviation is defined, authors should use the abbreviations and not a mix between the full word/expression and the abbreviation. Please correct.

Answer. SM was established as the abbreviation of silymarin and it was corrected along the document.

Introduction:

Line 51: remove the capital letter in “Reactive” (“…of Reactive oxygen species (ROS)…”), or add capital letter in the other 2 words. A consistency in the use of capital letters, when refereeing to chemicals, enzymes and others, should be maintained along the document (see lines 55-56, and 95, for example).

Answer. The capital letter R was removed from the manuscript, as in the other lines to which you refer.

Line 51: In the hydroxyl radical notation, the radical symbol should be placed near the oxygen (O) rather than near the hydrogen, please replace along the document.

Answer. The radical symbol was placed near oxygen as you recommended.

Line 54, in my opinion the authors should not abbreviate oxidative stress (OS), as it is not used many times and many non-common abbreviations difficult the reading.

Answer. The abbreviation OS in reference to oxidative stress was removed from the document.

Line 86: “NFE2L2 gene “, NFE2L2 needs to be in italic. Please use italic in all genes notations (revise all document).

Answer. All genes mentioned in the text were corrected to appear in italics.

Line 86. Please correct the dalton (Da) abbreviation, replace “66 kD protein” by “66 kDa protein”,

Answer. Dalton abbreviation was corrected to “66 kDa protein”.

Line 92.  Does “NF-E2” refers to NFE2, described in line 67? If so, use the same terminology/abbreviation along the document. If not, please define.

Answer. The complete name was defined of NF-E2. In the original paper,we refer to NF-E2 as nuclear factor erythroid 2, which is an erythroid-specific transcription factor.

Reference: 15.     Moi, P.; Chan, K.; Asunis, I.; Cao, A.; Kan, Y.W. Isolation of NF-E2-related factor 2 (Nrf2), a NF-E2-like basic leucine zipper transcriptional activator that binds to the tandem NF-E2/AP1 repeat of the beta-globin locus control region. Proceedings of the National Academy of Sciences of the United States of America 1994, 91, 9926-9930, doi:10.1073/pnas.91.21.9926.

In line 105, authors mention “Neh4 and Neh5 facilitate Nrf2 transcription”. This statement is not scientifically correct, as Nrf2 is a protein, and the term transcription defines a step of  DNA based gene expression, it is used to genes. There is some confusion of terminology. Please revise, you may use the following references (doi:10.1042/BJ20061611).

Answer. The statement was corrected, and “Nrf2 transcription” was removed from the text.

Section 2, (line 82 to 108) would benefit of a figure or scheme showing the domain architecture of Nrf2 and its interactions.

Answer. A figure describing Nrf2 interactions was added as you kindly suggested.

Line 110, do the authors mean “Under low stress cellular conditions” or under homeostatic conditions? In homeostatic conditions there’s a basal level of oxidative stress. Please refine the terminology. The Nrf2 proteosomal degradation occurs under basal conditions.

Answer. The phrase “Under low stress cellular conditions” was replaced by “under homeostatic conditions”.

Line 110, replace “little” by “low”

Answer. The word “little” was changed to “low” as you kindly suggested.

Line 118, please replace 26s by 26S.

Answer.  26s was replaced by 26S.

Line 118 – 121, the sentence seems incomplete.

Answer. The sentence was corrected.

Line 122 and line 152, please be consistent in the use of small musculoaponeurotic fibrosarcoma proteins abbreviation (sMaf and Maf are used). In Figure authors use sMaf. Please mention if Figure is original, or adapted, in case of adapted add the references.

Answer. We used the abbreviation sMaf for “small musculo-aponeurotic fibrosarcoma proteins” in the line tom which you refer and in Figure 1.

Line 205. I would suggest using “reactive nitrogen species (RNS)” instead of ”nitrogen reactive species (NRS)”. This would be consistent with the expression “reactive oxygen species”.

Answer. The term “nitrogen reactive species” (NRS) was replaced by “reactive nitrogen species” (RNS) as you suggested.

Line 212: authors refer to the “deletion” of CuZnSOD1 enzyme. The term deletion is generally used to genes or to amino-acids. Please refine the sentence.

Answer. The word “deletion” was removed and the information was re-organized.

Line 217 – Please use the correct notation of the gene mSod2 (knock out KO, refers to a gene), and this work refers to a mice gene not a human gene, please see original Ref. Please correct the gene notations, rigorously. This one and in the next lines.

Answer. Tue correct notation was used for genes in the document.

Line 256. A Figure to illustrate section 3.2. Nrf2 and Inflammation, would enrich the manuscript. The same also applies to the next sections. Please, add some figures.

Answer. A figure was added to enrich this section.

Line 336. Please define NO in “Nrf2 NO mice”

Answer. Previously in line 227, NO is defined. Is it necessary to define it again every time is mentioned in the manuscript?

Line 342: Please define: mTOR, mTORC1 mTORC2

Answer. mTOR refers to the signaling pathway of mammalian target of Rapamycin, mTORC1 complex 1 (mTORC1), and mTOR complex 2 (mTORC2). The reference was added to the text. 

Reference: Saxton, R.A.; Sabatini, D.M. mTOR Signaling in Growth, Metabolism, and Disease. Cell 2017, 169, 361-371, doi:10.1016/j.cell.2017.03.035.

Figure 2. Please correct the Figure notations (e.g. replace silybinB by silybin B, as in legend).

Answer. Figure spaces of silybin B were corrected.

Line 468-469: The sentence is ambiguous, please clarify it. It is not clear if overheat destroys or not silybin.

Answer.  Under overheated conditions, silybin is modified. This point is clarified in the text.

Line 469, please define the solvents identified by DMSO, THF, DMF,

Answer. Dimethyl sulfoxide (DMSO), acetone, tetrahydrofuran (THF), and dimethylformamide (DMF) were clearly defined.

Line 470 – 476: It would be helpful if in Figure 2 silybin structure have the Carbons numbered. And authors could include an oxidized structure.

Answer. The Carbon numbers were add to Figure 2 and the oxidized silybin structure

Line  481. Although the original article reports peak time as Tmax, the capitalized T is commonly used to temperature and not to time (which is t). Thus authors should define this quantity in the manuscript.

Answer. Tmax was replaced by tmax.

Line 481-484: Authors should mention that the described stomach absorption is not of free molecules but of molecules incorporated in micelles. Please clarify this point.

Answer. According to the references, SM flavolignans possess poor miscibility with other lipids, limiting their capacity to be absorbed in the lipid-rich outer membrane of enterocytes in the small intestine. We completed the information and added the reference:

  1. Yu, J.N.; Zhu, Y.; Wang, L.; Peng, M.; Tong, S.S.; Cao, X.; Qiu, H.; Xu, X.M. Enhancement of oral bioavailability of the poorly water-soluble drug silybin by sodium cholate/phospholipid-mixed micelles. Acta pharmacologica Sinica 2010, 31, 759-764, doi:10.1038/aps.2010.55.
  2. Perez-Sanchez, A.; Cuyas, E.; Ruiz-Torres, V.; Agullo-Chazarra, L.; Verdura, S.; Gonzalez-Alvarez, I.; Bermejo, M.; Joven, J.; Micol, V.; Bosch-Barrera, J., et al. Intestinal Permeability Study of Clinically Relevant Formulations of Silibinin in Caco-2 Cell Monolayers. International journal of molecular sciences 2019, 20, doi:10.3390/ijms20071606.

Please correct the English grammar concerning the verbal tenses, the number, and some spelling error. Also correct kilogram (kg not Kg), and other units highlighted in the manuscript.

Answer. The English grammar was reviewed by an expert to improve grammar related to verb tenses and structure. Also, Kg was corrected to kg.

Reviewer 2 Report

This review manuscript describes NRF2 and the NRF2 bioactivators such as Silymarin and flavolignans in two broad sections-1) basics on NRF2, NRF2 related signaling pathway, its role in various diseases 2) NRF2 bioactivators particularly Silymarin and flavolignans and their potential as a therapeutic molecule.

Overall, this review lacks a clear flow, it is lengthy yet not comprehensive, which makes it hard to follow. The title and the content seem to absolute mismatch with more than half a portion of the review describing NRF2 and NRF2 related signaling pathways. There are already excellent reviews recently published about the NRF2 signaling pathway. The contents of the Flavolignans /Sylimarins as NRF2 bioactivators are more descriptive rather than providing persuasive critical analysis and perspectives. Moreover, this manuscript requires a thorough revision by a native English-speaking scientist as the language throughout is lacking scientific rigors.

Author Response

We are very pleased with the results of the evaluation conducted by the Scientific Committee of your renowned journal of our manuscript entitled “Flavolignans from Sylimarin as Nrf2 Bioactivators and Their Therapeutic Applications”, and have carried out the task of responding to the well-targeted and greatly appreciated observations provided by the reviewers.

In this document, please find, in yellow, the changes kindly suggested by the Reviewers, which now appear as changes in the new version of the manuscript. Many thanks.

We follow here with the corrections and suggestions that we carried out on the article, taking into account the reviewer commentary:

Reviewer 2

This review manuscript describes NRF2 and the NRF2 bioactivators such as Silymarin and flavolignans in two broad sections-1) basics on NRF2, NRF2 related signaling pathway, its role in various diseases 2) NRF2 bioactivators particularly Silymarin and flavolignans and their potential as a therapeutic molecule.

Overall, this review lacks a clear flow, it is lengthy yet not comprehensive, which makes it hard to follow. The title and the content seem to absolute mismatch with more than half a portion of the review describing NRF2 and NRF2 related signaling pathways. There are already excellent reviews recently published about the NRF2 signaling pathway. The contents of the Flavolignans /Sylimarins as NRF2 bioactivators are more descriptive rather than providing persuasive critical analysis and perspectives. Moreover, this manuscript requires a thorough revision by a native English-speaking scientist as the language throughout is lacking scientific rigors.

We appreciate your comments regarded to this manuscript. In summary, the paper was corrected in different manners:

  1. We made corrections that reviewer 1 did about structure and format.
  2. Abbreviations and terminology were unified along the document.
  3. Some figures were added in different sections of the manuscript to enrich the content.
  4. According with reviewer 1 comment, some references were added to clarify specific points.
  5. As a general point, we wish to mention that the observation on the editing of the article in English has been punctiliously attended to, and a new review of the language of the entire manuscript has been carried out in order for this to be adequate.

Round 2

Reviewer 1 Report

The authors have clarified the required points and have improved the manuscript which now qualifies for publication.

After reading the revised version, I still would like to ask the authors to make some minor corrections.

Minor corrections:

Line 52. The radical symbol, please, on the left side of the O.

Line 230, please correct (Nrf2NO) to (Nrf2KO). The knock out abbreviation should be KO, and not NO. Please, correct also in lines 348 (no need to define here again), 353, 386, 393, 401.

Line 777, please replace OS by oxidative stress.

Author Response

Minor corrections

Line 52. The radical symbol, please, on the left side of the O.

Answer: Corrections were made. they are noted in yellow in the manuscript. Thank you

Line 230, please correct (Nrf2NO) to (Nrf2KO). The knock out abbreviation should be KO, and not NO. Please, correct also in lines 348 (no need to define here again), 353, 386, 393, 401.

Answer. Thanks for the observation. Changes were made and are in yellow in the manuscript.

Line 777, please replace OS by oxidative stress.

Answer. Thanks for the observation. Changes were made and are in yellow in the manuscript.

Reviewer 2 Report

In this revised version of the manuscript, the authors appear to hastily resubmit the manuscript without considering and addressing the recommended reviewers’ important comments. This revised manuscript is lengthy, yet not comprehensive particularly in the section describing the Flavolignans /Sylimarins as NRF2 bio activators. This section does not provide persuasive critical analysis and perspectives. Moreover, the authors spend 10 pages (out of 14 pages) on describing NRF2, NRF2- related signaling pathway and their role in various human diseases. This is unnecessary, considering already available many recent reviews on NRF2 signaling pathway. Moreover, the review has become exceptionally lengthy, mismatching the review title.

Author Response

Dea Dr. Yajun Li

Assistant Editor

Biomedicines

I do hope this letter finds you well. I would like to communicate to you that, since 2009, our investigation group has published the fruits of our research in your prestigious publishing house, MDPI, including 25 articles in: International Journal of Molecular Sciences (7), Molecules (6), Toxins (2), Antioxidants (2), Behavioral Sciences (1), Brain Sciences (1), Foods (1),materials (1) Cells (1), and Nutrients (3), for which we thank you so much for your support for publishing our work.

Indeed, one of our review articles has been multicited in JCR and in Scopus: Int. J. Mol. Sci. 2011, 12(5), 3117-3132; doi:10.3390/ijms12053117. Review Inflammation, Oxidative Stress, and Obesity. In addition, this review article is found among the 10 most frequently cited articles in this same journal (Int. J. Mol. Sci). In 2015, the review also won the "IJMS 2015's Best Paper Award"

Additionally, I have participated as a peer reviewer for 33 articles in MDPI, such as International Journal of Environmental Research and Public Health, Animals, Cells, Molecules, Biomolecules, Medicina, International Journal of Molecular Sciences, Marine Drugs, Materials, Nutrients.

Additionally, I have participated as Guest Editor Special Issue "Role for Antioxidants in Chronic Degenerative Diseases and Oxidative Stress" in Medical Sciences

I am writing to you because I am very concerned about an article that I sent for publication to the journal Biomedicines, and we are not in agreement with the verdict on it. Thus, I respectfully request your intervention in this unfortunate event.

In a very respectful way, we request for your help in order to understad the comments of the second reviewer. I briefly explain:

  1. In this revised version of the manuscript, the authors appear to hastily resubmit the manuscript without considering and addressing the recommended reviewers’ important comments.

Answer: We respond to the reviewer´s suggestions at the ten-day indicated by the Journal. It was not a hasty response as the reviewer says. Moreover, we respond to all the reviewer´s suggestions. Every reviewer´s comment was responded. The reviewer’s one comments and suggestions were responded point by point to more than thirty, for instance. For the reviewer’s two comments we point it out all the changes we made to improve the article.

  1. This revised manuscript is lengthy, yet not comprehensive particularly in the section describing the Flavolignans /Sylimarins as NRF2 bio activators.

Answer. Regarding this point the first reviewer´s opinión is different: “In this manuscript, the authors made an exhaustive and comprehensive review of the effects of flavolignans, from Sylimarin, as modulators of the Nrf2 signaling pathway.”

     So please, Dr. Yajun Li can you tell us which reviewer is right?, because on the same point they think different. We agree with reviewer one, and what about you?

  1. This section does not provide persuasive critical analysis and perspectives.

Answer. We consider in the section 5.3. Nrf2 Activated by Silymarin and Flavolignans: Promising Therapeutic Model the most recent scientific studies related to siymarin´s role in Nrf2 pathway. Therefore we designed the table 1 to resume the most relevant information related to it. The perspectives are described in the Conlution section at the end.

  1. Moreover, the authors spend 10 pages (out of 14 pages) on describing NRF2, NRF2- related signaling pathway and their role in various human diseases.

Answer. The prupose of this paper is describe the regulation of silymarin on NRF2 in several diseases and the therapeutic applications are analized. This is the reason to spend on describing NRF2 pathway in around 10 pages.

  1. This is unnecessary, considering already available many recent reviews on NRF2 signaling pathway.

Answer. We consider this is a comment from the reviewer´s point of view. If you take a look into Pubmed and research for Diabetes or Cancer there will be many original or review articles related to those topics, however, it does not neccessarily mean that everthing is already said We would appreciate if the reviewer could indicate us puntualy wich articles he means.

  1. Moreover, the review has become exceptionally lengthy, mismatching the review title.

Answer. Dr. Yajun Li, is there an specific number of pages or words stablished by MDPI for a review´s paper?

In sumary, we have taken into consideration the recommendations of the reviewers and we have responded point by point. We are very greatful with reviewer one as he has help us with his commets and suggestions to improve the manuscript. Timely response was given to all his comments that reviewer made in specific manner.

Regarded to reviewer two, please Dr. Yajun Li to help us understand his comments and specifically what does it mean. If you do understand them, you can help us. Thank you .

best regards

M en C Nancy Vargas.Mendoza

Dr. José A. Morales-González